# Race, employment, and the pandemic: An exploration of covariate explanations of COVID-19 case fatality rate variance

Christopher Griffin[1]*, Ray Block Jr.[2], Justin D. Silverman[3], Jason Croad[4], Robert P. Lennon[4]

**1** Applied Research Laboratory, Pen State University, University Park, State College, PA, United States of America, **2** Departments of Political Science and African American Studies, Penn State University, University Park, State College, PA, United States of America, **3** College of Information Science and Technology, Penn State University, University Park, State College, PA, United States of America, **4** Department of Family and Community Medicine, Penn State College of Medicine, Hershey, PA, United States of America

☯ These authors contributed equally to this work.
* griffinch@psu.edu

**Data Availability Statement:** All relevant data are within the paper and its Supporting information files.

## Abstract

We derive a simple asymptotic approximation for the long-run case fatality rate of COVID-19 (alpha and delta variants) and show that these estimations are highly correlated to the interaction between US State median age and projected US unemployment rate (Adj. $r^2$ = 60%). We contrast this to the high level of correlation between point (instantaneous) estimates of per state case fatality rates and the interaction of median age, population density and current unemployment rates (Adj. $r^2$ = 50.2%). To determine whether this is caused by a "race effect," we then analyze unemployment, race, median age and population density across US states and show that adding the interaction of African American population and unemployment explains 53.5% of the variance in COVID case fatality rates for the alpha and delta variants when considering instantaneous case fatality rate. Interestingly, when the asymptotic case fatality rate is used, the dependence on the African American population disappears, which is consistent with the fact that in the long-run COVID does not discriminate on race, but may discriminate on access to medical care which is highly correlated to employment in the US. The results provide further evidence of the impact inequality can have on case fatality rates in COVID-19 and the impact complex social, health and economic factors can have on patient survival.

## Introduction

The dynamics of COVID-19 have been extensively studied since the beginning of the pandemic (see e.g., [1–6]). As with many natural systems, the dynamics of COVID-19 are complex with non-linear second order and higher effects playing a role in the dynamics of transmission and fatality [7–10]. In particular, factors such as age, race, population density, socioeconomic status, and income inequality are all associated with [11–15] COVID-19 mortality.

**Funding:** The author(s) received no specific funding for this work.

Unemployment alone was negatively associated with morbidity in the first wave of the 1918–19 Spanish Flu pandemic, [16] but with a few exceptions [17–19] has not been thoroughly explored in this manner for COVID-19.

In this paper, we show evidence that unemployment is correlated with case fatality rate (CFR) when taking median age into consideration, which partially contradicts results in [18]. We explain this contradiction by the fact that we are using bulk US state-level data, while [18] uses zip-code level data which is known to have substantially more noise in measurement. We also address the question of whether these correlations are caused by the known higher COVID-19 mortality rates of racial minorities in the US and show that by incorporating this data we can explain 53.5% of the variance in case fatality rates. We also construct an asymptotic estimate for the (true) case fatality rate and show that this is correlated to an interaction between age and projected unemployment. This is important since both CFR and unemployment rate are transient measures. Interestingly, when the asymptotic estimates are used, the dependence on race disappears (as it should) suggesting that COVID does not discriminate on race, but may discriminate on access to medical care which is highly correlated to employment in the US.

A rich scholarly tradition—one that spans the writings of Anna Julia Cooper [20, 21] and W. E. B. Du Bois (particularly [22, 23]) to the more recent research of C. L. R. James (e.g., [24, 25]) William Julius Wilson [26] and Sandy Darity (see [27, 28])—teaches us that an appreciation for the interconnections between race and class is foundational to the study of social inequality. It is therefore impossible for theoretically-grounded and empirically-rigorous research on "racial disparities" and health outcomes to ignore the role of class divisions [29–32]. We therefore join the growing body of "disparities" scholarship by taking race/class associations seriously, particularly as they pertain to the coronavirus pandemic. In many ways, COVID-19 has exacerbated pre-existing social inequalities, deepening the divisions between rich and poor, White and non-White, etc. [33, 34]. Focusing on unemployment as a key economic indicator [35, 36], we investigate the role that race and class play in explaining variations in COVID-19 fatality rates.

The main results of this paper are:

1. We derive a simple asymptotic approximation for the long-run case fatality rate of COVID-19 (alpha and delta variants).

2. We show that these estimations are highly correlated to the interaction between US State median age and projected US unemployment rate (Adj. $r^2$ = 60.0%).

3. We contrast this to the high level of correlation between point (instantaneous) estimates of per state case fatality rates and the interaction of median age, population density and current unemployment rates (Adj. $r^2$ = 50.2%).

4. We show that incorporating race into the above models explains 53.5% of the variance in COVID case fatality rates for the alpha and delta variants when considering instantaneous case fatality rate. However, the dependence on race disappears when considering asymptotic projections of unemployment and CFR.

## Materials and methods

We approach the problem of modeling case fatality ratio (CFR) using an asymptotic analysis of an underlying Susceptible-Infected-Recovered-Deceased (SIRD) model, which is not explicitly calculated. Analysis is then carried out over data from the fifty US States. Standard

epidemiologic compartment models show that over the course of a pandemic, the case fatality ratio decreases over time to a limiting value. Consider a compartmented epidemic model (e.g., an SIRD model). Let $I$ be the instantaneous infected population and $D$ be the deceased population. Further let $C$ be the instantaneous cumulative case load. In an S1 File model, $C \equiv I$. SIRD models are inherently non-linear, usually having no closed form solution. Using a Taylor approximation, we can linearize the right-hand-side of the differential equations [37]. This allows us to approximate a closed form solution. Assuming a constant population size then linearizing about the current state of the compartmented epidemic model, the linearized solutions have form:

$$C \sim C_\infty - \kappa e^{-\alpha(t-t^*)}$$
$$D \sim D_\infty - \delta e^{-\beta(t-t^*)}.$$

Here, $\kappa$ and $\delta$ adjust the case fatality ratio to account for the initial condition (current value) and $\alpha$ and $\beta$ are rate constants. The terms $C_\infty$ and $D_\infty$ are the limiting values of case load and the deceased population respectively. The simplicity of the solutions lies in the underlying linearization of the nonlinear SIRD dynamics and allows the derivation of a statistical model to be fit. Dividing the terms, the model for case fatality ratio is then:

$$r \sim \frac{1 + Be^{-\beta(t-t^*)}}{K + Ae^{-\alpha(t-t^*)}} + \epsilon, \tag{1}$$

where $\epsilon \sim N(0, \sigma)$ is a noise variable accounting for experimental error. Here, $A = -\kappa/D_\infty$, $B = -\delta/D_\infty$ and $r_\infty = 1/K$ is the long-run case fatality rate. We note: starting from an SIR model would create a more complex form for these expressions. Since it's unclear whether COVID-19 caseload is following any simple SIR dynamic, we use the more parsimonious model. In the case of COVID-19, the measured caseload is believed to be a fraction of the true caseload. That is:

$$C_{\text{True}} = \tau \cdot C.$$

As $t \rightarrow \infty$, we may assume $\tau$ is relatively stable (constant) and thus the approximated case fatality rate is too large by a factor of $\tau$.

The choice of $t^*$ in Eq (1) is open. By varying $t^*$ over a number of possibilities and varying the size of the data tail we choose, we can construct a family (ensemble) of models following Eq (1) and use (i) the distribution on the computed $r_\infty$ to create upper and lower bounds on the estimator and (ii) use the various models to construct upper and lower bound predictors on the future behavior of CFR. For this study, we varied $t^*$ from 150 (days from January 22, 2020) to 330 (days from January 22, 2020) in increments of 10 days. The tail of the data used to fit Eq (1) started at day $t^*$ and proceeded to day 430, the last day of the data sample used.

Predicting the final CFR from early data is challenging, as there are confounding variables that may cause the model to over- or underestimate the true CFR [38]. For example, early models of COVID-19 did not account for the very high number of undiagnosed persons with COVID-19, leading to significant over-estimation of CFR. Further, different states may be at different points in the epidemic, leading to variances between states in CFR estimates. However, regardless of the ultimate CFR, the limiting behavior of CFR remains useful in analyzing factors that contribute to COVID-19 fatalities, because while different states may be at different places in their epidemic, the limiting behaviors should be invariant if we make two assumptions.

We use Adjusted-$r^2$ (rather than $p$-values) as one of our primary measures of goodness of fit. The goal of $r^2$ in this context is to assess whether specific predictors or explanatory variables

have a significant effect on the dependent variable. Many of our Adjusted-$r^2$ values fall between 0.5 and 0.6. Ozili [39] discusses acceptable $r^2$ values concluding that in social science, with its erratic human-behavior data, $r^2$ values between 0.1 and 0.5 are considered adequate provided some/most of the explanatory variables are statistically significant, $r^2$ between 0.5 and 0.99 are considered quite good. This is reflected in clinical data, where studies with $0.2 < r^2 < 0.4$ have been accepted [40]. Even traditional regression analysis would consider an $0.5 < r^2 < 0.7$ to have a moderate effect size [41] and Henseler et al. [42] proposed general guidance that $r^2$ of 0.5 was moderate (compared to $r^2$ of 0.75, which was considered substantial and $r^2$ of 0.25, which was considered weak).

## Case fatality rate modeling

In comparing CFRs between states, it is reasonable to ignore the effects of the undiagnosed over short time frames, because the population percentage of undiagnosed can be considered constant over recent history. Analysis of variations in states' CFR supports this. To compute instantaneous case fatality rate, we use data from Johns Hopkins COVID Dashboard (https://github.com/CSSEGISandData/COVID-19) from April 8, 2021. We then compare forecasts to data from October, 2020 and December 2020. We use these dates because after this time, vaccination was widely underway in the US and the less severe [43] Omicron variant would be responsible for further waves, thus distorting the underlying data on the alpha and delta variants.

At time $t$, the instantaneous case fatality rate was computed as:

$$r_{\text{now}} = \frac{D_t}{I_{t-14}},$$

where $D_t$ is the total number of deaths in a given state by time $t$ and $I_{t-15}$ is the total number of cases in that state at time assuming that recovery or death occurs within 14 days of diagnosis. Examining individual state case fatality rates shows substantial variation.

Fig 1 shows case fatality rates for 6 randomly selected states representing different regions of the country along with the fits of Eq (1).

We also assume that $\tau$ is related to the proportion of the population tested, denoted $p$, within each state. Here:

$$p = \frac{\text{test}}{\text{state population}}.$$

To assess the possibility that $\tau$ is not consistent across states, we regress $r_{\text{now}}$ against $p$. This is shown in Fig 2.

There is not significant correlation between the per-capita testing and instantaneous (computed) CFR, suggesting that at the point the data were taken, $\tau$ should be constant across all states. As we will see, this will support our hypothesis that variance among CFR (and asymptotic CFR) is explained by external social factors like unemployment and minority proportion.

It is also possible to use the per-state asymptotic CFR to estimate the asymptotic US CFR assuming that as $t \to \infty$ the entire population will be exposed to COVID19 and considering a no-vaccine scenario. We use the weighted average:

$$r_\infty^{\text{US}} = \frac{1}{P} \sum_s P^s r_\infty^s,$$

where $P$ is the total US population, $P^s$ is the population of state $s$ and $r_\infty^s$ is the asymptotic CFR of state $s$. Using this we obtain a 100% orders-statistic confidence interval on the CFR of

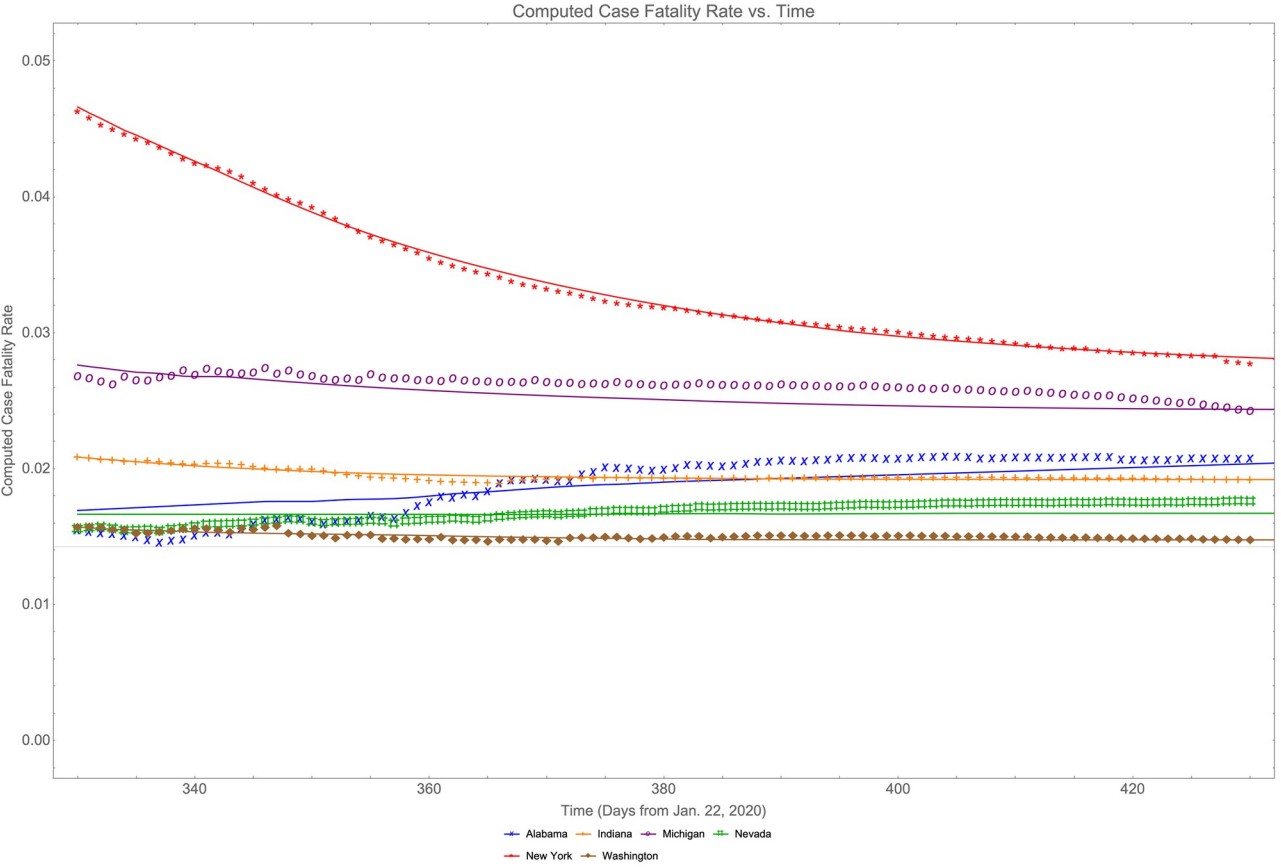

**Fig 1. Asymptotic behavior of case fatality rates for various states.**

(0.008, 0.019) with a pointwise estimator of 0.016. This is consistent with the single fit of cumulative US data, which estimates a nation-wide CFR of 0.0184, which is on the high-end of the confidence region. This is illustrated in Fig 3.

Verity et al., estimate the true 95% confidence interval on CFR as (0.0039, 0.0133) [38]. Comparing our CI with theirs, we see there is statistical agreement in the two approaches. Using Verity et al.'s model, we can estimate the under-sampling rate $\tau$ for the whole U.S. is contained in the interval (1.40, 4.76), which is consistent with past estimates.

Using the correction to $r_\infty$ we can compare the asymptotic case fatality rate to the instantaneous case fatality rate using the data from October 20, 2020. This is shown in Fig 4.

In this data, four states: Alabama, Kansas, Oklahoma, and Vermont, have asymptotic case fatality rates that are more than 1.5 times lower than their current estimated case fatality rate. However, if we apply the correction factors derived above, we see that all of those states may already be explained by unidentified cases. This is shown in Fig 5 where we show the region in which these states would lie if we correct using the estimate of $\tau$ derived for the U.S.

## Results

### CFR and unemployment

If we use the instantaneous (computed) case fatality rate over all states, 50.2% of the variance in this variable can be explained by the interaction of three simple variables: current

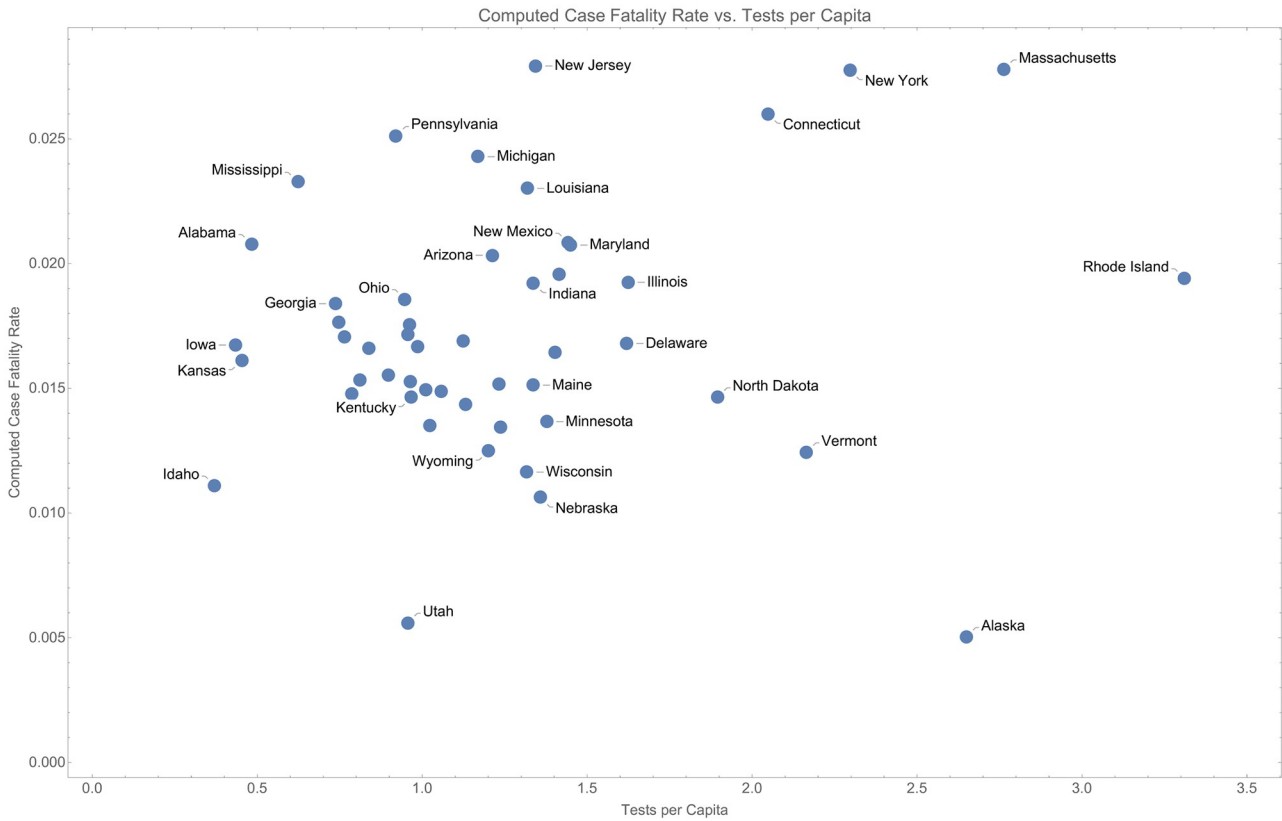

**Fig 2. Comparison of the instantaneous case fatality rate and proportion of people tested in each state shows no correlation.**

unemployment rate [44] ($u$), population density [45] ($d$) and median age [45] ($a$) with the model:

$$r \sim \gamma au + \beta \log_{10}(d) + \epsilon \qquad (2)$$

We use the log of the population because it generally improves adjusted $r^2$ while simultaneously decreasing AIC. The unemployment rate is provided by the US Dept. of Labor and reflects the unemployment recorded on March, 2021. (Descriptive statistics for all data used in this paper can be found in the S1 File). The parameter table for the model with significance is shown in below.

|  | Estimate | Standard Error | t-Statistic | P-Value |
|---|---|---|---|---|
| 1 | 0.0035 | 0.00198 | 1.744 | 0.0876 |
| $au$ | 0.00003 | $8.221 \times 10^{-6}$ | 3.6539 | 0.00065 |
| $\log(d)$ | 0.0039 | 0.0009 | 4.119 | 0.00015 |

On its own, unemployment is correlated with instantaneous case fatality rate ($p < 0.001$) and explains 27% of the variance. The model in Eq (2) along with the correlation of unemployment and instantaneous case fatality rate is shown in Fig 6. We tested a model in which we predict unemployment using the inverse of median age, population density and computed case fatality rate. This model explains only 26.9% (adjusted $r^2$) of the variance and the population

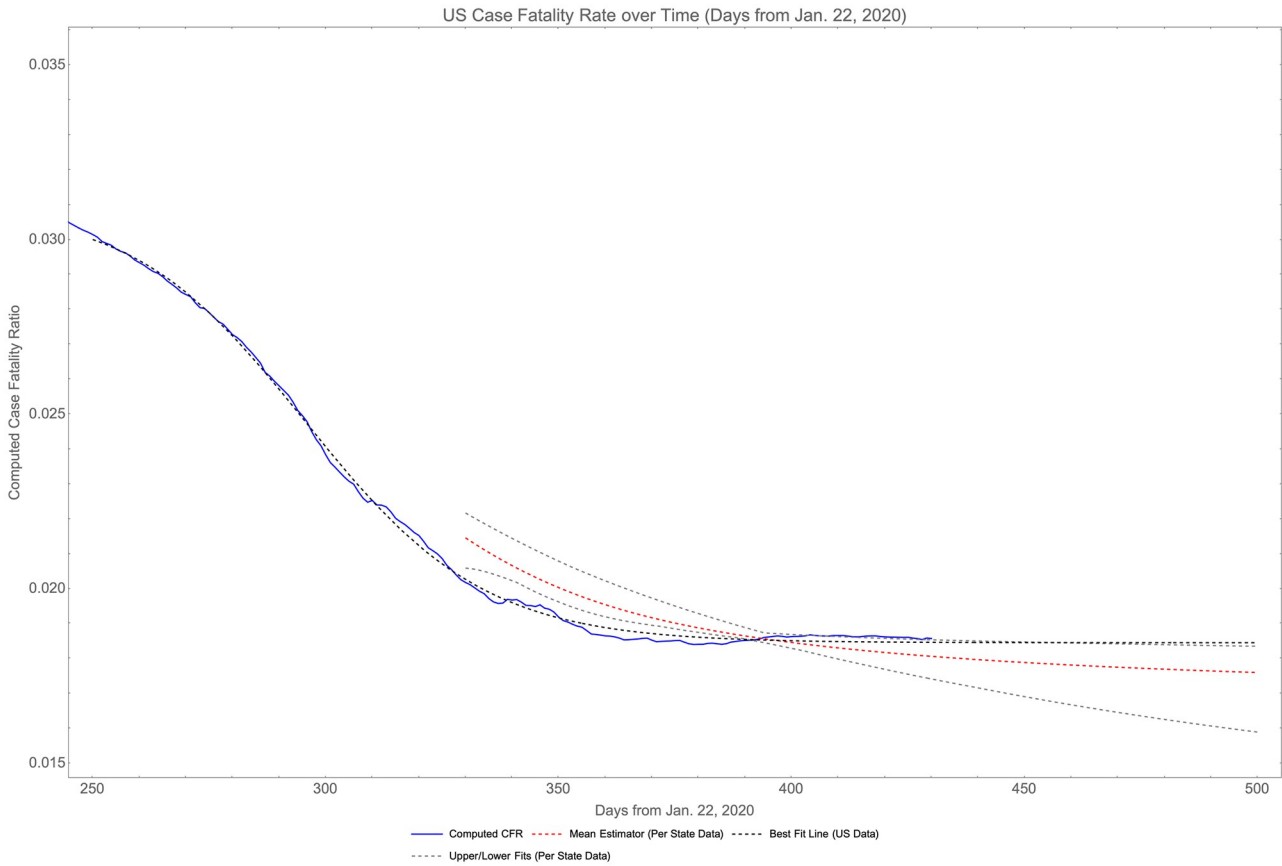

**Fig 3. Case fatality rate for the US as a whole through April 2021 with various multiples.**

density is not a significant estimator of unemployment as would be expected if Eq (2) could be used to solve for *u*. This suggests that we are not just seeing a simple correlation.

## Temporal and asymptotic analysis

The correlation between unemployment and instantaneous (computed) case fatality rate is consistent over time. If we use monthly unemployment data and the instantaneous CFR computed on (or about) the 20[th] of each month, then we can see that the correlation between instantaneous CFR and monthly unemployment were consistent since May 2020 (see Table 4 in the S1 File). The Bonferroni corrected p-value for all fits is $p = 0.0498$, showing that all these correlations are significant simultaneously at the 0.05 level. These results suggests a strong correlation between CFR and the existence of a correlation between structural unemployment in a state and its asymptotic CFR.

When the case fatality rates are corrected using asymptotic approximation and we replace *r* with $r_\infty$ in Eq (2), then 52.2% of the variance in $r_\infty$ is explained. However, current unemployment is a transient, while $r_\infty$ is used to measure long-run behavior. Supported by the correlation between instantaneous CFR and monthly unemployment illustrated in Table 4 in the S1 File, we construct an asymptotic projection for unemployment using a power-law:

$$u_t = c + \frac{a}{t^d}. \tag{3}$$

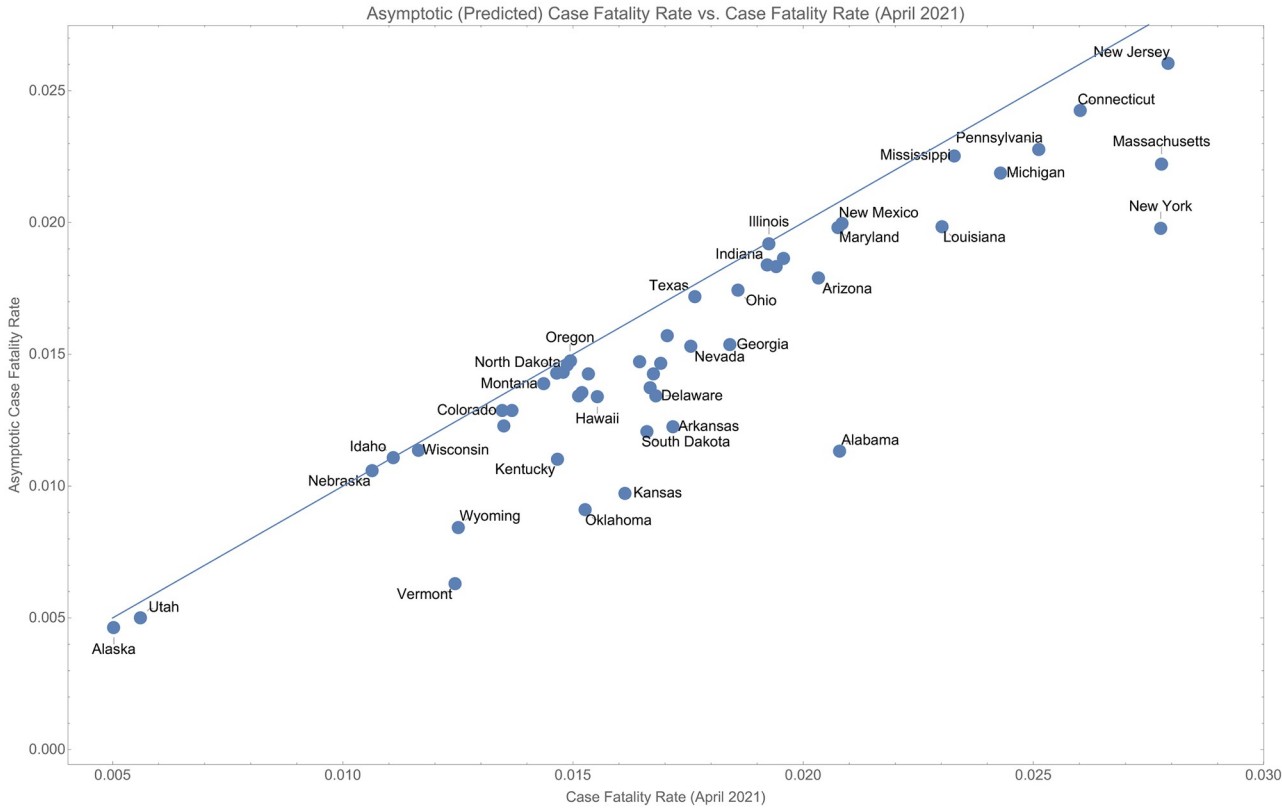

**Fig 4. Comparison of the true case fatality rate (ending April 2021) and the projected case fatality rate.**

This projection is constrained so that $c$ is not allowed to fall below unemployment from August 2019 [44], which was a point of historically low unemployment. As time goes to infinity, we can compute $u_\infty$. The relationship between unemployment in August 2019 and the projected unemployment is shown in Fig 7. Most states were projected to return to their pre-COVID unemployment levels. However, a small set of states may have longer term higher levels of unemployment as a result of COVID.

When we replace $u$ in Eq (2) with $u_\infty$, and adjust Eq (2) to be:

$$r_\infty \sim \gamma a u_\infty + \beta \log_{10}(d) + \epsilon,$$

we obtain the fit table:

|  | Estimate | Standard Error | t-Statistic | P-Value |
|---|---|---|---|---|
| 1 | 0.0004 | 0.0017 | 0.2405 | 0.8110 |
| $au_\infty$ | 0.00004 | $6.994 \times 10^{-6}$ | 5.6072 | $1.0528 \times 10^{-6}$ |
| $d$ | 0.0043 | 0.0007 | 5.76814 | $6.028 \times 10^{-7}$ |

On its own, the projected unemployment rate explains 27% of the variance in the asymptotic case fatality rate ($p < 0.001$) and again illustrates the correlation between case fatality and

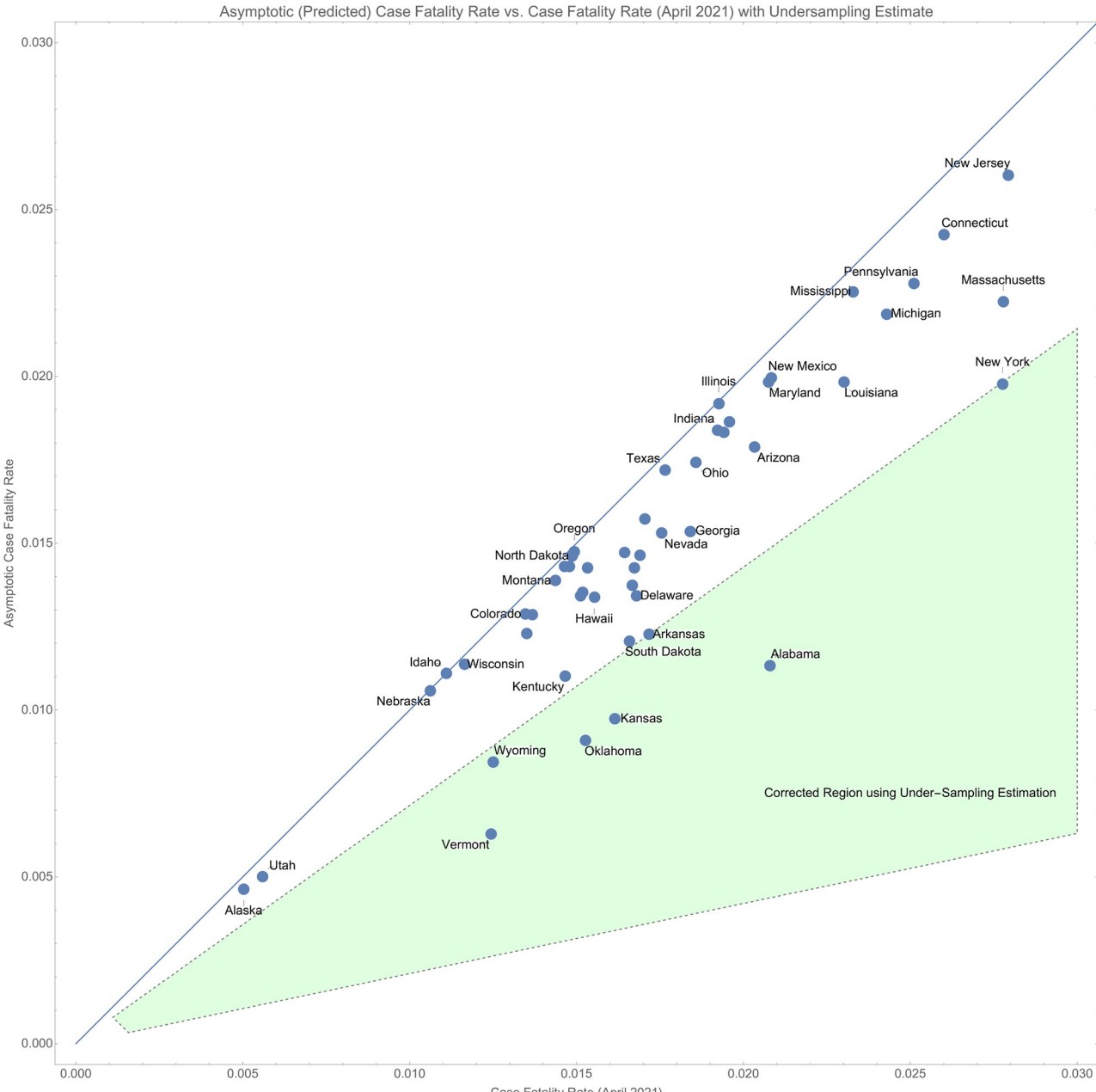

**Fig 5. The correction region showing the projection of the case rates forward agrees with the computed correction factor for the four outlier states.**

structural inequality and poverty. When interacting with median age and population density, 60.0% of the variance is explained. These fits are illustrated in Fig 8.

## CFR and race

There is a known relationship between CFR and minority status, with minority populations bearing the greater brunt of COVID-19. It is therefore reasonable to expect that the

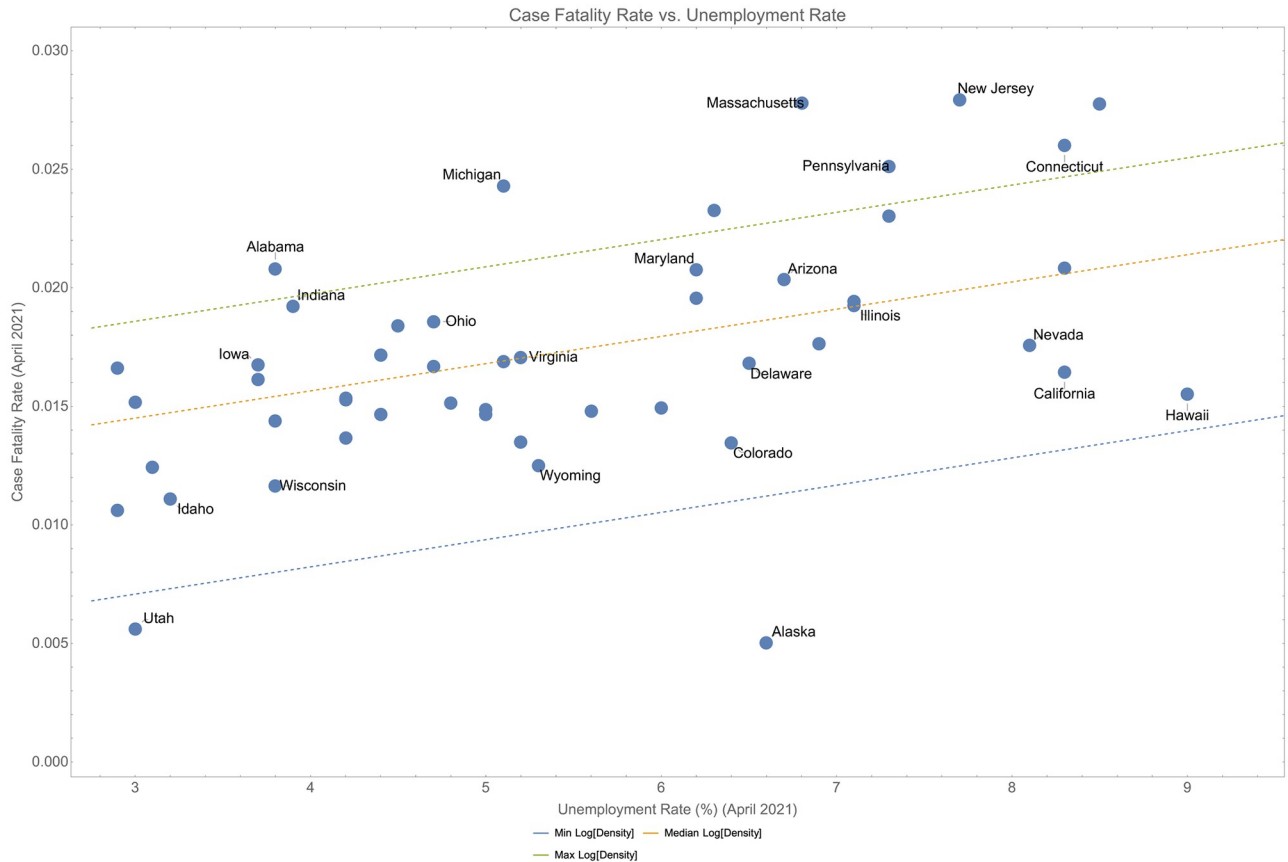

**Fig 6. Case fatality rate (April 2021) compared to unemployment rate across states and for various population densities.** The median of the median age is used for all plots.

relationship between unemployment and instantaneous CFR is a result of the higher rate of unemployment among minorities, which is illustrated in Fig 9 in the S1 File ($p < 0.001$).

We note that the correlation between minority population proportion and August 2019 unemployment is only weak ($p = 0.056$). This will be instructive when we evaluate the role minority status plays in asymptotic CFR. We now evaluate the relationship between minority population proportion and CFR using the simple linear model:

$$r \sim am + b,$$

where $m$ is the minority population proportion and as before $r$ is the CFR. Because there is substantial geographic diversity among minorities in the US, we perform 3 analyses: (i) an analysis of the relationship between minority proportion and instantaneous CFR in all 50 states, (ii) in the lower 48 states and (iii) in states where the population is composed of at most 30% minorities (total). Results of these analyses are shown in Tables 1 to 3.

The proportion of minorities (as a whole) is significantly correlated with instantaneous CFR only in the lower 48 states and the US as a whole. In low-minority states, it is not significantly correlated. However, the proportion of African Americans is always positively correlated with CFR. The remaining correlations (e.g., the negative correlation between Pacific Islander Americans and instantaneous CFR) are most likely spurious because of small sample size effects.

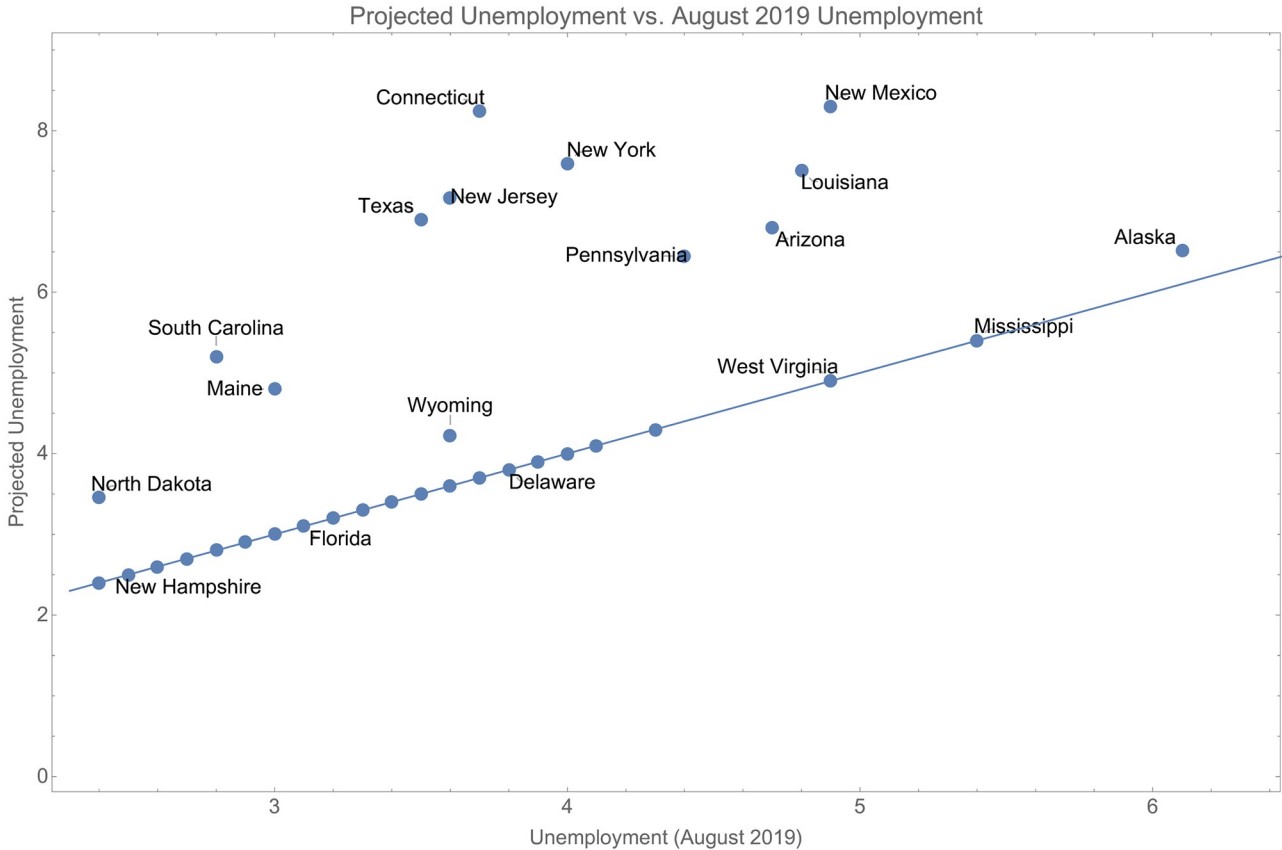

**Fig 7. The relationship between projected unemployment and unemployment in August 2019.**

These results suggest that comparing unemployment and the proportion of African Americans will help determine whether the relationship between unemployment and CFR is a minority population effect. The correlation between current unemployment and the proportion of African Americans in a state is not significantly correlated ($p = 0.259$) nor is the proportion of African Americans in a state significantly correlated to unemployment in August 2019 ($p = 0.12$), when the US was at or above full employment prior to the pandemic. Thus, it is unlikely that relationship between unemployment and CFR is a result of correlation between unemployment and minority proportion. This suggests a combined model.

## Combined model

When we alter Eq (2) to include an interaction term between the proportion of African Americans in a state and unemployment we have:

$$r \sim \gamma au + \beta \log_{10}(d) + aAu + \epsilon. \tag{4}$$

Here $A$ is the proportion of African Americans in the state. The resulting model parameters

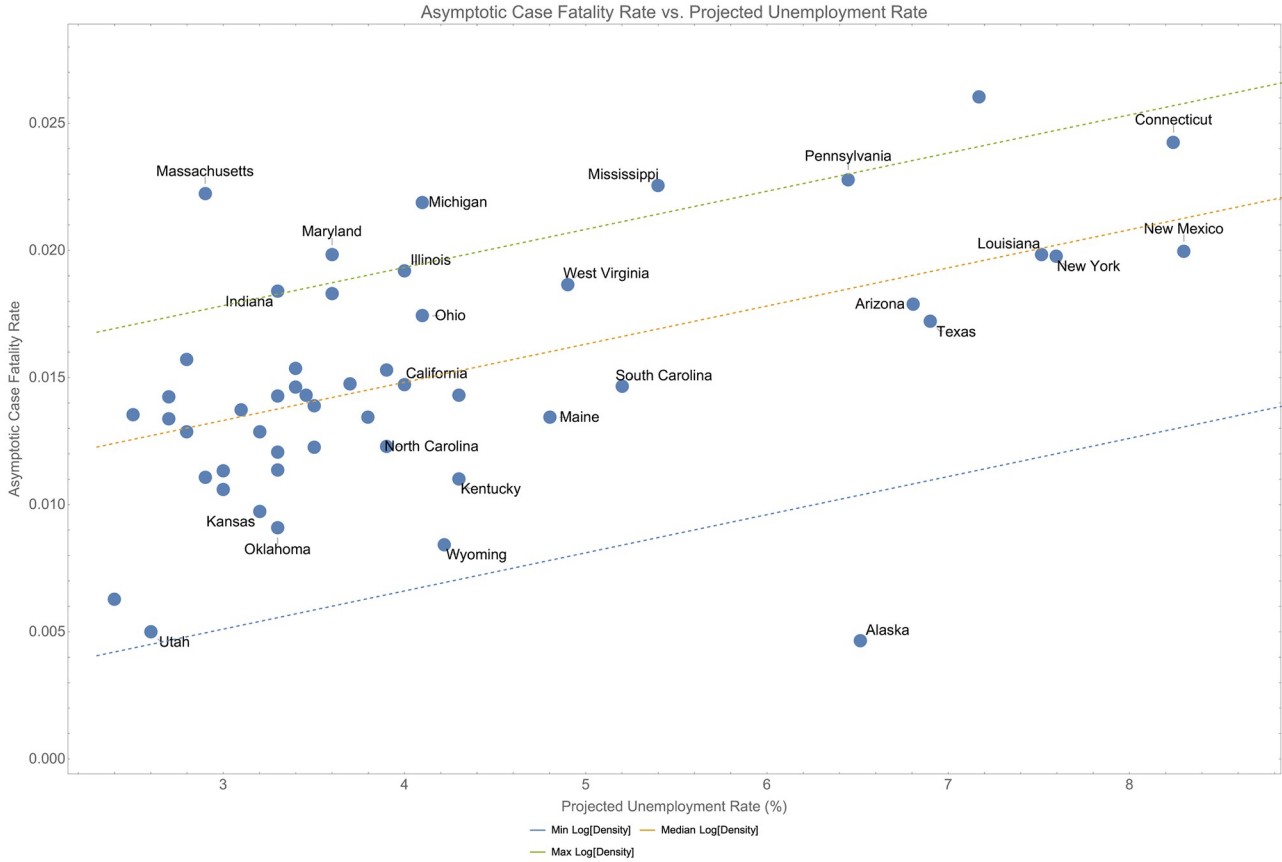

**Fig 8. When interacting with median age, asymptotic unemployment explains 60% of the variance in asymptotic case fatality rates.**

are all significant (see below).

|    | Estimate | Standard Error | t-Statistic | P-Value |
|----|----------|----------------|-------------|---------|
| 1  | 0.0045   | 0.00197874     | 2.263       | 0.0284  |
| $au$ | 0.00004 | $8.119 \times 10^{-6}$ | 3.2678 | 0.002   |
| $d$  | 0.0032   | 0.00098        | 3.15367     | 0.003   |
| $Au$ | 0.0020   | 0.00096        | 2.08461     | 0.043   |

**Table 1. Relationship between minority proportion and instantaneous CFR in all 50 states.**

| Group | Lin. Coeff. | p-val. of Corr. to CFR |
|-------|-------------|------------------------|
| African American | 0.0225 | 0.0017 |
| Latinx American | 0.0077 | 0.2641 |
| Asian American | 0.0066 | 0.6118 |
| Indigenous American | -0.0535 | 0.0250 |
| Islander American | -0.0471 | 0.3510 |
| Other American | 0.0455 | 0.0731 |
| Multi-Racial American | -0.02378 | 0.2858 |
| Non-Caucasian | 0.0079 | 0.0416 |

**Table 2. Relationship between minority proportion and instantaneous CFR in the lower 48 states.**

| Group | Lin. Coeff. | p-val. of Corr. to CFR |
|---|---|---|
| African American | 0.0207 | 0.0031 |
| Latinx American | 0.0065 | 0.3225 |
| Asian American | 0.0544 | 0.0294 |
| Indigenous American | -0.0262 | 0.3928 |
| Islander American | -1.0203 | 0.0059 |
| Other American | 0.0398 | 0.1037 |
| Multi-Racial American | -0.0365192 | 0.5651 |
| Non-Caucasian | 0.0107 | 0.0064 |

**Table 3. Relationship between minority proportion and instantaneous CFR where the population is composed of at most 30% minorities (total).**

| Group | Lin. Coeff. | p-val. of Corr. to CFR |
|---|---|---|
| African American | 0.0472 | 0.0081 |
| Latinx American | -0.0444 | 0.0586 |
| Asian American | 0.0665 | 0.4461 |
| Indigenous American | -0.0211 | 0.6116 |
| Islander American | -1.25 | 0.0065 |
| Other American | -0.1575 | 0.0454 |
| Multi-Racial American | -0.0707 | 0.6413 |
| Non-Caucasian | 0.0033 | 0.8126 |

This model explains 53.5% (Adjusted $r^2$) of the variation in the data and suggests that unemployment, age, population density and minority status interact in complex ways to increase the instantaneous CFR.

When we consider asymptotic CFR and modifying Eq (4) as:

$$r_\infty \sim \gamma a u_\infty + \beta \log_{10}(d) + a A u_\infty + \epsilon \tag{5}$$

no terms involving the proportion of African Americans in a state are significant (in Eq (5), $p = 0.485$ for the coefficient of $Au\infty$). Even replacing $Au_\infty$ with $A$ is not significant. This is sensible since in the long-run, COVID is not a disease that discriminates on race. It seems to be a disease that discriminates on access to medical care required to treat acute pneumonia that may occur as a result of infection.

## Discussion

The correlation between COVID-19 case fatality rate and both race and unemployment is consistent with the prevailing theory that minority disparity in COVID-19 outcomes stems from systemic racial injustice [46] which drives disparity across all sociodemographic covariates. This theory is further corroborated by analysis which has shown that mortality for persons able to access hospital care does not differ between African American and White patients after adjusting for sociodemographic factors and comorbidities [47], that racial disparities in COVID-19 appear to be driven primarily by unequal infection risks (i.e., sociodemographics) rather than case fatality rates [48], and that racial segregation explains higher African American mortality to such a degree that no as-yet unmeasured confounder is within the range of plausible covariates [14, 49]. Our model is also supported by the current (December 7, 2021)

CFR of 0.016 [50] which was our point-wise estimation of CFR from data through March 2021. The pointwise estimator as of March 2022 is 0.012 [50], reflecting improved vaccinations, treatments and the decreased aggressiveness of the Omicron variant [51]. However, this value is still well within our forecast confidence interval. In states with minority populations <30%, case fatality rates remain correlated with unemployment, but not with race. This suggests that unemployment alone is an independent risk factor for COVID-19, and may be considered in evaluating individual and group risk of COVID-19. Further, in populations with lower minority proportions, unemployment is a greater risk factor than race.

It is important that these findings not be misconstrued to suggest that racial inequity does not exist in states with low minority populations. Rather, it should be taken as evidence that, when it comes to social stratification, race and class are two sides of the same coin [43].

## Conclusion

In this paper we analyzed the interactions of the complex social phenomena of age, race, unemployment, population density and COVID-19 case fatality rate using both pointwise time series data and an asymptotic projection of unemployment and case fatality rate. We showed correlations to non-linear interactions between age, unemployment and age and race to COVID-19 CFR. We also showed that in the limit as time goes to infinity, we expect COVID-19 CFR to uncorrelate to race, which is consistent with the fact that COVID-19 fatality does not discriminate by race but may discriminate by access to medical care, which is predicted by socio-economic status in the US. This work is contrasted with the work in [18], which did not find correlation of CFR to unemployment but analyzed zip-code level data, which may have had higher noise levels in the unemployment rates. Future work should determine whether the observations made here can be translated to zip-code level data and further determine the impact of socio-economic status and race on COVID-19 fatalities.

More generally, our research speaks to the need for careful race/class analyses, for this need becomes especially apparent if one considers the pandemic's disproportionately adverse impact on both "health-" and "wealth-" related outcomes in African American communities [52–58]. As we note in the introduction, unequal outcomes across all aspects of the COVID-19 pandemic have highlighted systemic racial injustice in our society, and are appropriately driving calls for systemic change. The present paper offers data-driven support for such change, but also that unemployment is an independent, persistent risk factor for harm from the COVID-19 pandemic. Achieving equitable healthcare in the U.S. requires overcoming racial and economic disparities. Our work underscores the idea that the disparate impacts of COVID-19 must be both understood and addressed within an explicit framework of how race and class structure myriad arenas of American public life.

## Supporting information

**S1 Data.**
(ZIP)

**S1 File.**
(PDF)

## Author Contributions

**Conceptualization:** Christopher Griffin, Ray Block, Jr., Justin D. Silverman, Jason Croad, Robert P. Lennon.

**Data curation:** Christopher Griffin.

**Formal analysis:** Christopher Griffin, Justin D. Silverman.

**Methodology:** Justin D. Silverman.

**Writing – original draft:** Christopher Griffin, Ray Block, Jr., Jason Croad, Robert P. Lennon.

**Writing – review & editing:** Christopher Griffin, Ray Block, Jr., Justin D. Silverman, Jason Croad, Robert P. Lennon.

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
