## [Decision Letter · Decision Letter 0]

18 Jul 2022

PONE-D-22-10078Race, employment, and COVID-19: an exploration of covariate explanations of COVID-19 case fatality rate variancePLOS ONE

Dear Dr. Griffin,

Thank you for submitting your manuscript to PLOS ONE. After careful consideration, we feel that it has merit but does not fully meet PLOS ONE’s publication criteria as it currently stands. Therefore, we invite you to submit a revised version of the manuscript that addresses the points raised during the review process.

We look forward to receiving your revised manuscript.

Kind regards,

Emrah Gecili, PhD

Academic Editor

PLOS ONE

Journal Requirements:

Reviewers' comments:

Reviewer's Responses to Questions

**Comments to the Author**

1. Is the manuscript technically sound, and do the data support the conclusions?

Reviewer #1: Yes

Reviewer #2: Yes

2. Has the statistical analysis been performed appropriately and rigorously? 

Reviewer #1: Yes

Reviewer #2: No

3. Have the authors made all data underlying the findings in their manuscript fully available?

Reviewer #1: Yes

Reviewer #2: Yes

4. Is the manuscript presented in an intelligible fashion and written in standard English?

Reviewer #1: Yes

Reviewer #2: Yes

5. Review Comments to the Author

Reviewer #1: This is an interesting study and the author addressed everything perfectly. In this study, the author explained the models and showed the results with online data. Overall, this is interesting but may be a little more introduction and discussion would be more helpful.

Reviewer #2: Please define terms. ex: \\kappa, \\delta ect.

Please explain why you assume "linearized forms of solutions". May be useful for readers who aren't familiar with this model.

Adj. $r^2 \\approx 50-60 \\%$ is not that good. Please justify or use other criteria too if possible.

Since data are available, provide a reproducible code if possible.

Report descriptive statistics of data for the variables you used.

6. PLOS authors have the option to publish the peer review history of their article (what does this mean?). If published, this will include your full peer review and any attached files.

Reviewer #1: **Yes: **Mohammad Bhuiyan

Reviewer #2: No

---

## [Author Response · Author response to Decision Letter 0]

29 Jul 2022

Please see the response to reviewers file included in the submission. We have addressed each reviewer comment within that document.

---

## [Decision Letter · Decision Letter 1]

30 Aug 2022

Race, employment, and the pandemic: an exploration of covariate explanations of COVID-19 case fatality rate variance

PONE-D-22-10078R1

Dear Dr. Griffin,

We’re pleased to inform you that your manuscript has been judged scientifically suitable for publication and will be formally accepted for publication once it meets all outstanding technical requirements.

Kind regards,

Emrah Gecili, PhD

Academic Editor

PLOS ONE

Additional Editor Comments (optional):

Reviewers' comments:

Reviewer's Responses to Questions

**Comments to the Author**

1. If the authors have adequately addressed your comments raised in a previous round of review and you feel that this manuscript is now acceptable for publication, you may indicate that here to bypass the “Comments to the Author” section, enter your conflict of interest statement in the “Confidential to Editor” section, and submit your "Accept" recommendation.

Reviewer #1: All comments have been addressed

Reviewer #2: All comments have been addressed

2. Is the manuscript technically sound, and do the data support the conclusions?

Reviewer #1: Yes

Reviewer #2: Yes

3. Has the statistical analysis been performed appropriately and rigorously? 

Reviewer #1: Yes

Reviewer #2: Yes

4. Have the authors made all data underlying the findings in their manuscript fully available?

Reviewer #1: Yes

Reviewer #2: Yes

5. Is the manuscript presented in an intelligible fashion and written in standard English?

Reviewer #1: Yes

Reviewer #2: Yes

6. Review Comments to the Author

Reviewer #1: Please improve the resolution of the figures. they are hard to understand and very vague. make them 300 dpi.

Reviewer #2: (No Response)

7. PLOS authors have the option to publish the peer review history of their article (what does this mean?). If published, this will include your full peer review and any attached files.

Reviewer #1: **Yes: **Mohammad Bhuiyan

Reviewer #2: No

---

## [Editor Report · Acceptance letter]

5 Sep 2022

PONE-D-22-10078R1 

Race, employment, and the pandemic: an exploration of covariate explanations of COVID-19 case fatality rate variance 

Dear Dr. Griffin:

I'm pleased to inform you that your manuscript has been deemed suitable for publication in PLOS ONE. Congratulations! Your manuscript is now with our production department. 

Kind regards, 

on behalf of

Dr. Emrah Gecili 

Academic Editor

PLOS ONE